# Application of CRISPR-Cas9 System to Study Biological Barriers to Drug Delivery

**DOI:** 10.3390/pharmaceutics14050894

**Published:** 2022-04-20

**Authors:** Ji He, Riya Biswas, Piyush Bugde, Jiawei Li, Dong-Xu Liu, Yan Li

**Affiliations:** 1School of Science, Auckland University of Technology, Auckland 1010, New Zealand; ji.he@aut.ac.nz (J.H.); riya.biswas@aut.ac.nz (R.B.); piyushbugde@gmail.com (P.B.); jiawei.li@aut.ac.nz (J.L.); dong-xu.liu@aut.ac.nz (D.-X.L.); 2The Centre for Biomedical and Chemical Sciences, School of Science, Faculty of Health and Environmental Sciences, Auckland University of Technology, Auckland 1010, New Zealand; 3School of Interprofessional Health Studies, Auckland University of Technology, Auckland 1010, New Zealand

**Keywords:** CRISPR-Cas9, blood-brain barrier, intestinal epithelial barrier, drug permeability

## Abstract

In recent years, sequence-specific clustered regularly interspaced short palindromic repeats (CRISPR)-CRISPR-associated (Cas) systems have been widely used in genome editing of various cell types and organisms. The most developed and broadly used CRISPR-Cas system, CRISPR-Cas9, has benefited from the proof-of-principle studies for a better understanding of the function of genes associated with drug absorption and disposition. Genome-scale CRISPR-Cas9 knockout (KO) screen study also facilitates the identification of novel genes in which loss alters drug permeability across biological membranes and thus modulates the efficacy and safety of drugs. Compared with conventional heterogeneous expression models or other genome editing technologies, CRISPR-Cas9 gene manipulation techniques possess significant advantages, including ease of design, cost-effectiveness, greater on-target DNA cleavage activity and multiplexing capabilities, which makes it possible to study the interactions between membrane proteins and drugs more accurately and efficiently. However, many mechanistic questions and challenges regarding CRISPR-Cas9 gene editing are yet to be addressed, ranging from off-target effects to large-scale genetic alterations. In this review, an overview of the mechanisms of CRISPR-Cas9 in mammalian genome editing will be introduced, as well as the application of CRISPR-Cas9 in studying the barriers to drug delivery.

## 1. Introduction

Accumulating evidence has suggested membrane transporter proteins play pivotal roles in drug permeability across biological membranes and thus determine the efficacy and safety of drugs [1,2,3,4,5,6]. Two major transporter superfamilies, namely the ATP-binding cassette (ABC) and the solute carrier (SLC) transporters, have been identified to modulate the drug absorption and disposition. The human ABC transporter family contains 48 members with 7 subfamilies, including several important drug transporters, such as P-gp (ABCB1), MRP2 (ABCC2) and BCRP (ABCG2). They are widely distributed in various tissues, functioning to actively extrude pharmacologically diverse substrate drugs out of cells against the concentration and chemical-potential gradients by using energy derived from ATP hydrolysis. The SLC transporters include 298 influx transporters responsible for nutrient intake and drug disposition into various organs. Certain ABC and SLC transporters are co-localised in a specific tissue and play in concert with selective permeability to specific drugs, excluding circulating drugs and toxic agents but allowing essential nutrients and other drugs into the tissue. Using endogenous transport pathways may lead to more effective drug delivery into the pharmacological sanctuaries. However, the contribution of the transporter(s) to overall drug permeability across a biological membrane is often unknown or cannot be accurately measured. Although the heterogeneous expression systems have been used as valuable tools, the interpretation of results could be complicated by the presence of endogenous transporters and species differences. With the development of CRISPR-Cas9 genome editing techniques, it is now possible to efficiently study the functional consequences of genetic mutations and delineate the interactions between membrane proteins and drugs.

This review elaborates on the principles and application of CRISPR-Cas9 genome editing techniques with a special focus on drug permeability-related membrane proteins.

## 2. CRISPR-Cas9 System

The CRISPR-Cas adaptive immune systems are a natural defence mechanism of bacteria against foreign genetic elements [7]. Some types of these CRISPR-Cas systems have been repurposed to facilitate precise genome engineering in eukaryotic cells [8,9]. Generally, CRISPR-Cas systems are categorised into two classes (1 and 2), which are further subdivided into six types (I–VI) based on the structure and function of Cas protein [10,11]. Class 1 CRISPR-Cas systems (type I, III and IV) recruit multi-subunit effectors, in contrast to the single effector of class 2 (type II, V and VI) [10,11]. CRISPR-Cas9 comprises three major components, Cas9 nuclease, CRISPR RNA (crRNA) and trans-activating crRNA (tracrRNA).

The type II CRISPR-SpyCas9 derived from *Streptococcus pyogenes* is one of the best developed and broadly used systems in site-specific genetic engineering of human cells [8,12,13]. SpyCas9 has a bi-lobed architecture formed by a recognition (REC) lobe and a nuclease (NUC) lobe (Figure 1) [14]. An arginine-rich (R-rich) Bridge helix (BH) connects these two lobes [15]. In the case of genetic engineering of human cells using the CRISPR-SpyCas9 system, either the hybridised crRNA-tracrRNA duplex combining a 42-nt crRNA and an 80-nt tracrRNA or a modified single 102-nt guide RNA (sgRNA, crRNA fused to tracrRNA) is loaded onto SpyCas9 to form ribonucleoprotein (RNP) and direct RNP to the target site bearing a 5′-NGG-3′ PAM (N = A/T/G/C) [14,15,16,17,18]. Notably, the 10-nt PAM-proximal seed region is critical for the SpyCas9-catalysed DNA cleavage. Some mismatches may be tolerated in the rest of the gRNA sequence [15].

The NUC lobe of SpyCas9 contains the HNH and RuvC nuclease domains, which cleave the target and non-target strands of the target DNA through the single-metal and two-metal mechanisms, respectively [14,15,19,20,21]. SpyCas9 nuclease thus stimulates a DSB at the target locus, i.e., normally 3-nt upstream PAM. These DSBs are mainly re-ligated by one of the two major DNA repair pathways, the error-prone non-homologous end joining (NHEJ) and the high-fidelity homology-directed repair (HDR) in mammalian cells [22]. NHEJ mediates gene knockouts (KOs) through the formation of indel (nucleotides insertion/deletion) known to facilitate frameshift mutations and premature stop codons [23]. Dual-gRNA-induced multiple DSBs can additionally facilitate more extensive excisions in the genome [24]. By contrast, the activation of HDR is contingent on the cell type and state, since it is normally active in dividing cells [25]. HDR can typically knock in a pre-designed repair template to the open reading frame (ORF) to terminate the transcription of the target gene. The integrated constructs may comprise genes encoding biomarkers for screening and probing purposes, such as antibiotic-resistant genes and genes encoding fluorescent probes.

Aside from the canonical SpyCas9 from *Streptococcus pyogenes*, many other Cas9 orthologs and the type V CRISPR-Cas12 systems have been uncovered and used for mammalian genome editing, as summarised in Table 1 [16,26,27,28,29,30,31,32,33,34,35,36,37,38,39,40].

The common negative effects of CRISPR-Cas9 include the endonuclease activity-induced cell damage and the off-target effect. The recently developed techniques, such as base editing and prime editing, may help minimise the off-target effect of CRISPR-Cas9 [41,42,43]. The cleavage efficiency of CRISPR-Cas9 could be influenced by factors in terms of designing sgRNA and Cas9 constructs (e.g., construction and composition of sgRNA and Cas9 protein) [33,44], the GC content, secondary structure and nucleotide preference of sgRNA [44,45], and the primary target sequence [46]. However, emerging evidence suggests that some more complicated matters may also impact CRISPR-Cas9’s editing efficiency, such as different chromatin states (i.e., euchromatin and heterochromatin) [47], the location of target DNA in the nucleosome [25,47,48], truncated protein isoforms [47,49], induction of DNA damage responses (e.g., *p53* and *KRAS*) [50,51,52], and large-scale gene rearrangements [23].

Chromatin states can affect both Cas9 binding and the repair pathway choice. The relatively unpacked euchromatin is more accessible to Cas9 protein over heterochromatin [47]. It was also reported that the frequencies of long double-stranded donor-based HDR were higher at heterochromatin compared to euchromatin. In contrast, NHEJ frequencies were higher in euchromatin [25]. Furthermore, Smits, Ziebell [49] reported that residual protein expression was found in about one third out of 193 KOs at variable levels from low to original. These truncated protein isoforms could remain with their original cellular functions and presumably involve other unknown roles [53].

Moreover, the Cas9-induced DSBs were found to cause p53-dependent cellular toxicity and eventually reduce cellular viability [50,52]. Similar to p53, the wild-type *KRAS* gene might hamper the growth of KO cells [51]. These findings support the involvement of *p53* and *KRAS* in CRISPR-Cas9-induced DNA damage response (DDR) activation, leading to the selective advantage of the *p53*- and *KRAS*-mutant cells. In addition, large-scale gene rearrangements, chromosomal translocations, gene inversions or large insertions/deletions were reported in a comprehensive study of Cas9-induced mutagenesis [23]. Similarly, chromosome structural alterations, such as micronuclei and chromosome bridges, were observed in mouse embryos after CRISPR-Cas9 genome editing [54]. These genomic alterations may induce ectopic expression of other genes.

Despite the potential drawbacks that the CRISPR-Cas9 system has, it is superior to other genome editing technologies, such as zinc-finger nucleases (ZFNs) and transcription activator-like effector nucleases (TALENs), considering its ease of design, cost-effectiveness, greater on-target DNA cleavage activity, multiplexing capabilities, and wide suitability for diverse cell types and organisms [8,55]. Moreover, as compared to gene knockdown technologies, such as RNAi (i.e., siRNA and shRNA), CRISPR-Cas9 is thought to present advantages, including lower off-target activity, stable and heritable complete elimination of the target gene expression and multiple genome editing potentials [56,57]. These advantages make CRISPR-Cas9 a reliable and versatile gene-editing approach and enable advances in molecular biology research for a variety of applications.

## 3. Application of CRISPR-Cas9 in Drug Delivery Barrier Studies

### 3.1. Intestinal Barriers to Drug Delivery

In order for orally administered drugs to exert their beneficial effects (other than on the GI tract itself), they must be delivered to their target organ(s) and tissues by systemic circulation. Adequate concentrations at the site(s) of action are only achieved by overcoming several absorption and metabolism barriers, both in the intestine and in the liver. Atypical absorption kinetics of many drugs suggest their intestinal absorption cannot be simply predicted from their physicochemical properties, and their interactions with intestinal ABC and SLC transporters may lead to limited or nonlinear intestinal permeability and absorption of drugs, resulting in extensive variability in their oral bioavailability and inadequate plasma concentrations and lack of pharmacological effect.

P-gp (MDR1/ABCB1) is a 170-KDa efflux transporter located on the plasma membrane in many tissues, such as the intestine, liver, kidney and brain. It exerts a critical barrier role in the intestinal absorption of lipophilic and amphipathic drugs with diverse pharmacological actions. Indeed, the oral bioavailability of its substrate talinolol in humans can be increased by 34% when co-administered with the P-gp inhibitor, erythromycin [58]. Similarly, coadministration of oral Cys A enhanced the human oral bioavailability of paclitaxel and docetaxel by 7- and 10-fold, respectively, and reduced interpatient variability in the systemic exposure of docetaxel to that seen in intravenous administration [59,60]. CRISPR-Cas9 gene editing in cell culture and animal models for drug transport has mainly focused on *ABCB1*.

#### 3.1.1. Knockout of Abcb1 in MDCK Cells by CRISPR-Cas9

The main in vitro models of intestinal absorption include subcellular fractions, cell cultures, isolated tissues and membrane vesicles. In recent years, cultured cells, such as Caco-2 or Madin-Darby canine kidney (MDCK) cells, have been increasingly used to study drug absorption.

Madin-Darby canine kidney II cells (MDCK) heterogeneously expressing single or multiple human transport proteins are commonly used models to study polarised drug transport and identify substrates. However, endogenous canine transporters such as canine Mdr1/P-glycoprotein (Abcb1) and canine Mrp2 (Abcc2) transport various drugs and complicate the interpretation of directional transport studies [61]. Complete KO of endogenous canine *Abcb1* (*cAbcb1*) (homozygous disruption) (2) resulted in indistinguishable differences in directional transport of model human ABCB1 substrates, such as digoxin, labetalol and quinidine [61,62]. Similarly, a comparison of efflux ratios (ER) between MDCKI wild-type and gMDCKI (KO of endogenous cAbcb1) showed that out of 135 compounds tested, 38% showed efflux activity in MDCKI wt, while no significant efflux was observed in gMDCKI cells [63]. Further overexpression of human ABCB1 (MDR1) in those *cAbcb1*-KO MDCK cells generated less complicated drug transport models, which enabled substrate identification and removed the interference from cAbcb1 (Figure 2) [62,63]. The *cAbcb1*-KO MDCK cells human overexpressing BCRP (ABCG2) were also generated and used to accurately identify human BCRP substrates, which may be potentially transported by cAbcb1 (Figure 2) [64]. The expression of BCRP and MRP2 transcripts and proteins in the human small intestine is similar to that of MDR1 [65,66], suggesting these two efflux transporters also play important roles in regulating the oral absorption of their substrates.

#### 3.1.2. Mdr1a/b Double-Knockout Rat Models

Drug permeability across the intestine can be easily extrapolated across mammalian species due to the similar composition of the epithelial cell membranes. Laboratory animals are commonly used to predict oral drug absorption in humans by determining absolute bioavailability with the comparison of area under the plasma concentration-time curve (AUC) after intravenous and oral administration. They are also used for the prediction of potential absorption-based drug interactions by comparing oral bioavailability values with and without coadministration of another drug. With the manipulations in embryonic stem cells commonly used to create mouse knockouts, transgenic mice have provided an appropriate model to investigate the roles of specific drug transporter(s) or metabolising enzyme(s). For example, intestinal P-gp limits the bioavailability of a wide range of drugs, including paclitaxel, digoxin, aliskiren, betrixaban, celiprolol, fexofenadine and talinolol, as shown in P-gp knockout mice [65,66,67]. However, species differences exist in gastrointestinal pH, intestinal flora mobility, transit time, and activity and expression level of transporters and metabolism enzymes. Although rats are empirically superior to mice as models for intestinal drug absorption, the generation of transgenic rat models was difficult, as rat genes are much more difficult to manipulate using embryonic stem cells. A novel MDR1 (Mdr1a/b) double-knockout (KO) rat model was generated by the CRISPR/Cas9 system without any off-target effect detected for compensatory mechanisms (e.g., CYP3A subfamily and transporter-related genes) [68]. The rate and extent of oral absorption of digoxin, a typical MDR1 substrate, was significantly increased in *Mdr1a/b* (-/-) rats compared with WT. With the high efficient KI and KO via CRISPR genome editing technologies, more useful in vivo tools (e.g., humanised animal models) would be generated for studying drug absorption barriers.

### 3.2. Biological Barriers to Anticancer Drugs

Many anticancer drugs are targeting intracellular DNA or proteins and thereby cause DNA damage or inhibition of DNA synthesis and inhibition of cell division. The intracellular concentration of some anticancer drugs is vital, and it has been suggested that the intracellular concentration of a drug is the product of a competition between its passive or active uptake rate and either an active efflux rate or metabolic rate. However, the mechanisms whereby some anticancer drugs enter cancer cells and overcome the biological barriers remain poorly understood. The CRISPR-Cas9 system has provided extra tools to unfold the underlying mechanisms at the genome-scale and simplified the interpretation of results.

#### 3.2.1. Knockout and Regulation of ABC Transporter Genes in Cancer Cells by CRISPR-Cas9

Although the clinical relevance between ABC transporters and multidrug-resistance (MDR) phenotype is controversial and unclear, extensive in vitro studies strongly support their potential roles in the cellular pharmacokinetics of substrate drugs [53,69]. The CRISPR-Cas9 gene manipulation technique has been used to reverse ABC transporter-related MDR and restore non-malignant phenotype in many types of cancer cell models. It was reported that KO of *ABCB1* significantly enhanced the sensitivity to doxorubicin (aka., Adriamycin (ADR)) in the ADR-resistant ovarian cancer cell line A2780/ADR [70], breast cancer cell line MCF7/ADR [71], osteosarcoma cell line KHOSR2 and U-2OSR2 [72]. Similarly, KO of *ABCB1* in two ABCB1-overexpressing multidrug-resistance (MDR) cell lines, KBV_200_ and HCT-8/V, remarkedly improved the sensitivity and accumulation of ABCB1 substrate drugs, such as vincristine and doxorubicin. Furthermore, the sensitivity of carfilzomib (CFZ)-resistant myeloma cell line AMO-CFZ [73] and acute lymphoblastic leukaemia (ALL) cell line HALO1 [74] to CFZ was recovered after knocking out *ABCB1* using the CRISPR-Cas9 system.

In addition, a CRISPR/Cas9 KO of BEN domain-containing protein 3 (BEND3) upregulated efflux transporter breast cancer resistance protein (BCRP; ABCG2) and reduced the intracellular levels of TAK-243 and induced resistance in acute myeloid leukaemia (AML) cells [75]. Similarly, silencing ubiquitin-editing enzymes A20 by CRISPR/Cas9 modulated brentuximab vedotin sensitivity (BV, a drug-conjugated anti-CD30 antibody) in Hodgkin lymphoma line L428, occurred through NF-kappaB-mediated ABCB1 expression [76]. Targeting NF-kappaB activity synergised well with BV in killing Hodgkin lymphoma cell lines, augmented BV sensitivity, and overcame BV resistance in vitro and in Hodgkin lymphoma xenograft mouse models.

Therefore, CRISPR-Cas9-mediated modulation of ABC transporter genes or their regulator(s) may represent a promising in vitro cancer cell model for substrate identification and molecular pathology research that potentially contributes to uncovering novel therapeutic biomarkers.

#### 3.2.2. Genome-Wide CRISPR-Cas9 Knockout Screen

Historically, genome-wide loss-of-function screening in mammalian cells has employed the RNA interference (RNAi) gene knockdown technology. However, this method is limited by incomplete protein depletion and off-target effects-induced false positives [77]. The application of CRISPR-Cas9 in genome-scale functional screening may overcome these drawbacks of RNAi and simplify the interpretation of the loss of gene function. GeCKO is the first library of sgRNAs targeting 5′ constitutive exons of 18,080 genes in the human genome with an average coverage of 3–4 sgRNAs per gene, which was applied to both negative and positive selection screens in human cells models [78]. The optimised sgRNA libraries, for the human and mouse genomes, named Brunello and Brie, respectively, have been created by maximising on-target activity and minimising off-target effects to enable more effective and efficient genetic screens [79]. Genome-wide CRISPR screens reveal that expression of the multidrug-resistant gene *ABCC1* and the lysosomal transporter SLC46A3 differentially impact tumour cell sensitivity to PCA062, a P-cadherin targeting antibody-drug (DM1, a maytansine-derived potent microtubule-inhibiting agent) conjugate for the treatment of multiple cancer types, including basal-like breast cancer [80]. ABCC1 confers antibody maytansine conjugate resistance [81], and SLC46A3 transports catabolite of antibody maytansine conjugate from lysosome to cytoplasm [82,83]. Silencing ABCC1 could lead to cytoplasmic accumulation of the warhead, while KO SLC46A3 cause lysosomal accumulation (Figure 3).

A genome-wide CRISPR/Cas9 knockout screen identified that SLC1A3 confers L-asparaginase resistance in human prostate cancer PC3 cells [84]. L-asparaginase serves as a crucial medicine for adolescent acute lymphoblastic leukaemia but is frequently associated with solid tumour resistance. ASNase stimulates aspartate and glutamate consumption and decreases their intracellular concentrations. SLC1A3 is an aspartate and glutamate transporter, and overexpression of SLC1A3 may promote cancer cell proliferation via “enhanced permeability” of aspartate and glutamate and fueling aspartate, glutamate and glutamine metabolisms. SLC1A3 inhibition caused cell cycle arrest or apoptosis and myriads of metabolic vulnerabilities in the tricarboxylic acid cycle, urea cycle, nucleotides biosynthesis, energy production, redox homeostasis and lipid biosynthesis.

#### 3.2.3. Novel Mechanisms of Platinum Accumulation in Cancer Cells

The platinum-containing drugs such as cisplatin, carboplatin and oxaliplatin have been widely integrated into the standard and preferred regimens for various solid cancers. Several SLC transporters, such as CTR1 (SLC31A1) and CTR2 (SLC31A2), have been reported to participate in cisplatin influx in several heterogeneous expression systems (e.g., yeast and mouse embryonic fibroblast models) [85]. However, complete KO of either *CTR1* or *CTR2* in ovarian carcinoma OVCAR8 cells showed indistinguishable differences in cisplatin sensitivity compared to wild-type cells [86]. A study using CRISPR-Cas9-mediated gene deletion suggests volume-regulated anion channels (VRACs), configured as leucine-rich repeat-containing 8 (LRRC8) heteromers, significantly contribute to cisplatin and carboplatin accumulation in human cells, accounting for 50–70% of total platinum accumulation under isotonic conditions [87]. Loss of LRRC8D causes resistance to carboplatin and cisplatin, but not to oxaliplatin in KBM-7 leukaemia cells [87]. To put this into perspective, silencing LRRC8D also confers cisplatin resistance in *BRCA1*-mutated ovarian cancer cells in a genome-scale CRISPR-Cas9 knockout screen study [88], and a retrospective analysis of two small cohorts of ovarian cancer patients that received platinum drugs reveals that lower expression levels of LRRC8D were correlated with lower survival rates [87]. Further research is warranted to establish greater confidence in supporting experimental and clinical correlative data and address key gaps in current knowledge.

### 3.3. Blood-Brain Barriers

Endothelial cells that line the microvasculature of the central nervous system (CNS) constitute the blood-brain barrier (BBB), selectively excluding circulating drugs and toxic agents from entering the majority of the central nervous system but allowing essential nutrients, hormones and certain drugs into the brain. Drug permeability across BBB is mainly determined by the interactions between their physicochemical characteristics and the specialised BBB features, including tight intercellular junctions that markedly limit paracellular permeability, plus a unique expression of ABC and SLC transporters that determine the transcellular permeability. The two most abundant ABC transporters in the human BBB are ABCB1 (MDR1) and ABCG2 (BCRP) [89], which function as pivotal rate-limiting barriers to drug distribution or access to the brain [90]. Moreover, one attractive approach to brain drug delivery is to utilise SLC transporters as an efficient vehicle to circumvent BBB [91]. CRISPR-Cas9 allows cost-effective gene manipulation at more physiologically relevant levels and readily expands our knowledge of the novel territories.

Solute carrier family 35, member F2 (SLC35F2), is highly expressed in the BBB and is localised exclusively on the apical membrane of brain microvascular endothelial cells (BMECs), differentiated from human induced pluripotent stem cells (hiPS-BMECs). Silencing SLC35F2, by CRISPR/Cas9 mediated knockout, diminished the apical-to-basolateral transport and intracellular accumulation of its substrate drug YM155 in hiPS-BMECs [92]. By contrast, in studies using an in situ brain perfusion in mice, neither CRISPR/Cas9 KO of Slc35f2 nor pharmacological perturbation reduced brain uptake of YM155. YM155 is a substrate of human and mouse SLC35F2 [92], and SLC35F2 is a major determinant of YM155 antitumour efficacy in xenografted models, comparing the effects of YM155 on tumour growth between wild-type and SLC35F2 KO SW480 cells [93]. It is suggested that the limited role of slc35f2 in the distribution of YM155 in mouse brain was due to a substantial uptake mediated by organic anion transporting polypeptide oatp1a4 [92].

OATP1A2 (SLC21A3) is abundantly expressed in the apical (blood) side of the human BBB [94,95,96] and facilitates the transport of analgesic opioids (e.g., [D-penicillamine2,5]encephalin, deltorphin II), antimigraine triptans (e.g., sumatriptan), levofloxacin and methotrexate [4]. A mouse line humanised for human OATP1A2 was established by CRISPR-Cas9 mediated knock-in of the coding region downstream of the mouse *Oatp1a4* promoter [97]. Such a humanised mouse model would be an invaluable tool for studying the transcellular transport of drug substrates across the blood-brain barrier (BBB). OATP1A2 mRNA in the brain was increased, corresponding to the disappearance of Oatp1a4, and OATP1A2 was localised on both the luminal and abluminal sides of the BBB. However, incomplete translation or posttranslational modification of OATP1A2 occurred in the BBB, as evidenced by the peptide-dependent quantitative levels of OATP1A2, leading to the lack of functional transport of model substrates across BBB. These examples highlight the fact that we are still a long way from being able to generate clinic-relevant BBB transporter models.

### 3.4. Regulation of Transporter Genes

The CRISPR/Cas9 system has been applied to edit non-transcribed DNA sequences, including DNA methylation tags. DNA methylation, occurring at the 5-carbon position of cytosine residues located in dinucleotide CpG sites and regions of high CpG density, called CpG islands (CGIs), has been observed in the promoter regions of 60–70% of genes in mammals, which are involved in the epigenetic regulation of genes [98]. ABCC3 is highly expressed in human skin tissues, with its mRNA accounting for 20% of the total mean transporter mRNA content [99]. The expression of ABCC3 mRNA showed large interindividual variability (9.5-fold) but cannot be explained by single nucleotide polymorphisms. In human skin HaCaT cells, the disruption of the region surrounding a CGI, located approximately 10 kb upstream of the ABCC3 gene, by CRISPR/Cas9 led to significantly decreased ABCC3 mRNA levels [99]. Consistently, ABCC3 mRNA was upregulated in HaCaT cells by the demethylating agent 5-aza-2′-deoxycytidine. A better understanding of epigenetic regulation of skin transporter genes is important for the design of drug delivery across this essential barrier.

Multidrug and toxin extrusion protein 1 (MATE1), which is encoded by solute carrier 47A1 (SLC47A1), functions as the final excretion step of drugs/toxins into bile and urine. Its substrates include vitamin thiamine, cimetidine, metformin, guanidine, procainamide, antiviral agents (e.g., acyclovir and ganciclovir) and antibiotics (e.g., cephalexin and cephradine) as well as an endogenous substrate creatinine [4]. Some differences in the pharmacokinetics of MATE1 substrate drugs cannot be explained by genetic variations in humans. MATE1 mRNA expression levels negatively correlated with methylation levels of the CpG island in the 27 kb upstream of *SLC47A1*. The CRISPR-Cas9-induced deletions in this CpG area significantly lower *SLC47A1*/*MATE1* mRNA expression in HepG2 cells. This study highlights the importance of epigenetic regulation in pharmacokinetics and the application of CRISPR/Cas9 for editing non-transcribed DNA targets in human cells.

## 4. Conclusions

The breathtaking CRISPR-Cas9-based gene-editing technology is undoubtedly an excellent genetic manipulation method for pharmacology and molecular biology studies. It provides researchers with a convenient tool to identify gene functions with significant advantages, including ease of design, cost-effectiveness, greater on-target DNA cleavage activity, and multiplexing capabilities. The exploitation of CRISPR-Cas9 techniques has expanded the tools to study biological barriers to drug delivery and generated novel insights in the field and may lead to the development of promising strategies to overcome therapeutic failures and minimise drug toxicity. The genome-scale CRISPR-Cas9 KO screen study also facilitates the identification of novel genes in which loss alters drug permeability across the biological membrane and thus, modulates the efficacy and safety of drugs. With the new strategies to enhance KI and KO efficiency via CRISPR genome editing technologies, more interpretable in vitro and in vivo tools are expected for studying barriers to drug delivery.

## Figures and Tables

**Figure 1 pharmaceutics-14-00894-f001:**
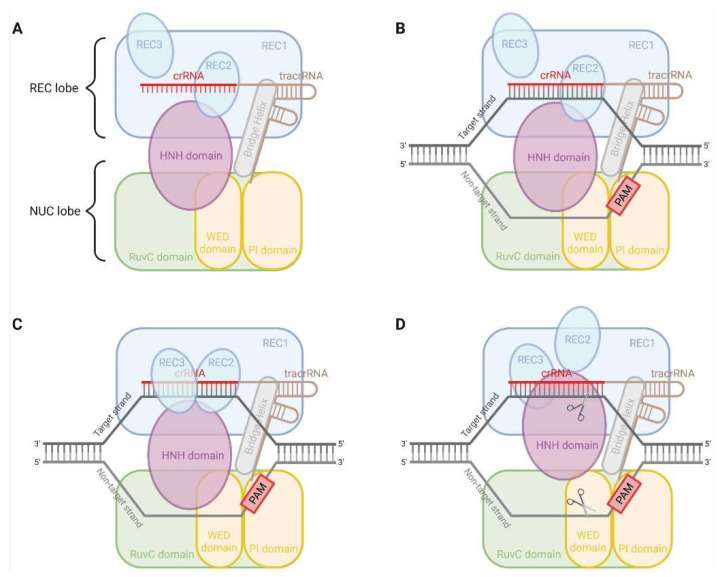
Schematic illustration of DNA recognition and cleavage by CRISPR-SpyCas9. (**A**) RNA duplex is loaded onto SpyCas9 to form ribonucleoprotein (RNP). (**B**) The PAM-interacting (PI) domain of the NUC lobe recognises 5′-NGG-3′ PAM and facilitates the binding of crRNA to the target DNA to form R-loop. (**C**) The REC3 domain of the REC lobe and Bridge helix (BH) sense mismatches. (**D**) The REC2 domain of the REC lobe undergoes a large outward rotation, leading to the conformational transition of the HNH domain into an active state. The HNH and RuvC domain of the NUC lobe then cleaves the target and non-target strands of the target DNA, respectively. The cleavage site is always located at 3- to 4-nt upstream of PAM. *Created with BioRender.com*.

**Figure 2 pharmaceutics-14-00894-f002:**
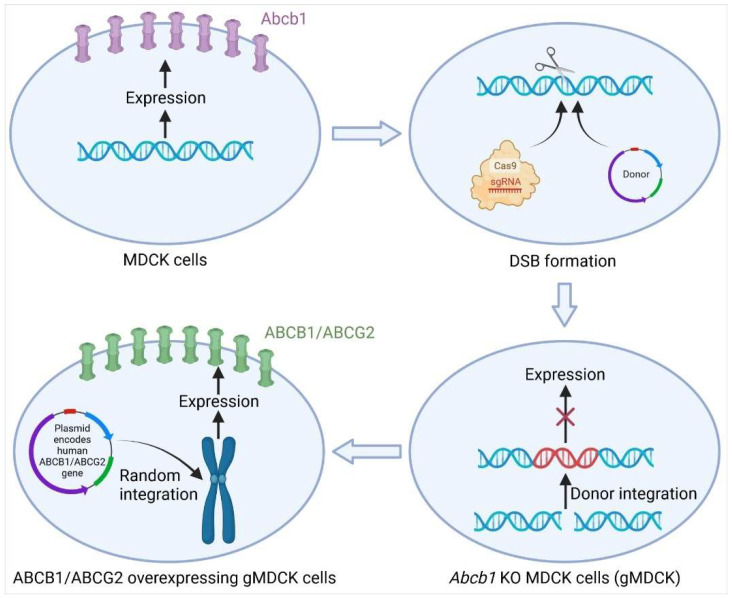
Schematic illustration of generating *Abcb1* KO MDCK (gMDCK) cells and ABCB1/ABCG2 overexpressing gMDCK cells. After transfection, the target sequence is recognised and cleaved by the sgRNA-Cas9 complex, followed by the integration of the donor template through the HDR repair pathway. The expression of *Abcb1* is thus disrupted in *Abcb1* KO MDCK (gMDCK) cells. To generate ABCB1/ABCG2 overexpressing gMDCK cells, plasmids encoding human *ABCB1* or *ABCG2* genes are delivered into gMDCK cells and randomly integrated into the genome, leading to constant overexpression of ABCB1 or ABCG2 protein. *Created with BioRender.com*.

**Figure 3 pharmaceutics-14-00894-f003:**
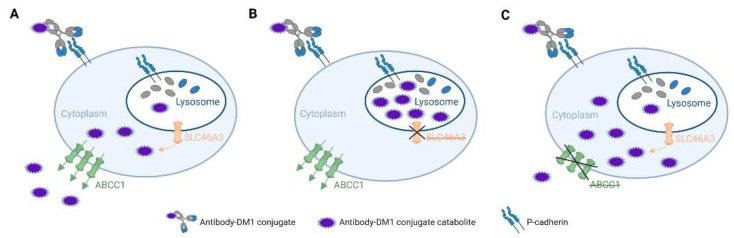
Proposed mechanisms of cellular distribution of catabolite of antibody maytansine conjugate in wild-type (**A**), SLC46A3-KO (**B**) and ABCC1-KO (**C**) cancer cells. *Created with BioRender.com*.

**Table 1 pharmaceutics-14-00894-t001:** Summary of CRISPR-Cas systems used for genome editing of mammalian cells.

Class 2	Subtype	Effector Nuclease	Size (aa)	Target	TracrRNA Requirement	Seed Sequence Requirement	PAM Sequence	Cleavage Product
**Type II**	A	SpyCas9	1368	dsDNA (or ssDNA/ssRNA with PAMmers)	Yes	Yes	NGG	DSB (blunt end)/SSB
	A	St1Cas9	1121	dsDNA	Yes	Yes	NNRGAA	DSB (blunt end)
	A	St3Cas9	1388	dsDNA	Yes	Yes	NGGNG	DSB (blunt end)
	A	SauCas9	1053	dsDNA/ssRNA	Yes	Yes	NNAGAAW/-	DSB (blunt end)/SSB
	B	FnoCas9	1629	dsDNA/ssRNA	Yes	Yes	NGG/-	DSB (blunt end)/SSB
	C	CjeCas9	984	dsDNA/ssRNA	Yes	Yes	NNNVRYM/-	DSB (blunt end)/SSB
	C	NmeCas9	1082	dsDNA/ssDNA	Yes/No	Yes	NNNNGATT/-	DSB (blunt end)/SSB
**Type V**	A	Cas12a	1200–1500	dsDNA/ssDNA	No	Yes	Optimal 5′ T-rich and suboptimal C-containing PAMs/-	DSB (sticky end with 5-nt 5′-overhang)/SSB
	B	Cas12b	1100–1300	dsDNA/ssDNA	Yes	Yes	Optimal 5′ T-rich and suboptimal C-containing PAMs/-	DSB (sticky end with 6-nt 5′-overhang)/SSB
	E	Cas12e	<1000	dsDNA	Yes	Unknown	5′ T-rich PAMs	DSB (sticky end with 10-nt 5′-overhang)
	F	Cas12f	400–600	dsDNA/ssDNA	Yes	Unknown	5′ T-rich PAMs/-	DSB (sticky end with 5’-overhang)

N represents A, T, G and C; V represents A, C, and G; M represents A and C; R represents A and G; W represents A and T; Y represents C and T.

## Data Availability

Not applicable.

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
