# Peer review of "Application of CRISPR-Cas9 System to Study Biological Barriers to Drug Delivery"

_pharmaceutics, 2022, doi:10.3390/pharmaceutics14050894_

Round 1
Reviewer 1 Report
The title of the review paper indicated that it was focused on biological barriers to drug delivery so it was expected that the review would be focused on various applications of CRISPR editing on understanding drug transport/drug delivery. However, the review is largely focused on other areas of CRISPR based gene editing and thus the title may be inappropriate. The majority of the text is occupied with a relatively comprehensive review of CRISPR-Cas9 systems, a section on factors influencing Cas9 editing, and then a comparatively short section on "drug delivery barrier studies" - largely focusing on ABCC1 studies in a few organisms. The review reads like several distinct, and not necessarily related sections, and this reviewer would suggest splitting this review into different separate and focused manuscripts. The section on the Cas9/Cas12a systems and the factors are interesting - especially the one on the review of editing factors - but they really dont have much to do with the last part.
The last section related to the Drugs - is the weakest of the sections - and is very incomplete with no coherent discussion of the various barriers to drug transport - e.g. intestine, brain, liver, etc - and some arbitrary selection of studies, largely single KO based studies. A more comprehensive survey of the literature is needed and consideration of the different drug transport barriers.
There are a few language issues that should be addressed.
Line 42 - important organs? what organ is unimportant?
Line 43 - in a specific tissue and play in concert with selective - needs rewording.
Line 46 - pharmacological sanctuary - unusual word choice, reword.
Line 508 -
Line 515 - In our practice, the vast majority of clones exhibited this kind of genotype. These results have been presented in 516 New Zealand Breast Cancer Symposium 2021 (NZBCS) (http://www.nzbcs.org.nz/).
This sentence was unclear as to what clones the author's are referring and the methods used and should be clarified.
Author Response
The title of the review paper indicated that it was focused on biological barriers to drug delivery so it was expected that the review would be focused on various applications of CRISPR editing on understanding drug transport/drug delivery. However, the review is largely focused on other areas of CRISPR based gene editing and thus the title may be inappropriate. The majority of the text is occupied with a relatively comprehensive review of CRISPR-Cas9 systems, a section on factors influencing Cas9 editing, and then a comparatively short section on "drug delivery barrier studies" - largely focusing on ABCC1 studies in a few organisms. The review reads like several distinct, and not necessarily related sections, and this reviewer would suggest splitting this review into different separate and focused manuscripts. The section on the Cas9/Cas12a systems and the factors are interesting - especially the one on the review of editing factors - but they really dont have much to do with the last part.
The last section related to the Drugs - is the weakest of the sections - and is very incomplete with no coherent discussion of the various barriers to drug transport - e.g. intestine, brain, liver, etc - and some arbitrary selection of studies, largely single KO based studies. A more comprehensive survey of the literature is needed and consideration of the different drug transport barriers.
Ans: Many thanks for the constructive comments. The review manuscript has been restructured to balance the basic science and its application in the studying drug transport barriers.
There are a few language issues that should be addressed.
Line 42 - important organs? what organ is unimportant?
Ans: The word “important” has been changed into “various”
Line 43 - in a specific tissue and play in concert with selective - needs rewording.
Ans: This has been changed into “play pivot roles in determining selective permeability….”
Line 46 - pharmacological sanctuary - unusual word choice, reword.
Ans: This has been changed into “pharmacological sanctuaries” The authors argue that the term is common in pharmacokinetics field.
Line 508 -
Line 515 - In our practice, the vast majority of clones exhibited this kind of genotype. These results have been presented in 516 New Zealand Breast Cancer Symposium 2021 (NZBCS) (http://www.nzbcs.org.nz/).
This sentence was unclear as to what clones the author's are referring and the methods used and should be clarified.
Ans: We appreciate the comments. These sentences have been removed in the restructured manuscript.
Reviewer 2 Report
Ji He and colleagues present a quality and well-written review manuscript describing application of CRISPR-Cas9 system to study biological barriers to drug delivery.
Authors provide a thorough overview of the mechanisms of the widely used CRISPR-Cas systems in mammalian genome editing, as well as the challenges that restrict editing efficiency and the application of CRISPR-Cas9 in studying barriers to drug delivery.
Authors suggest that CRISPR-Cas9 has benefited the proof-of-principle studies for a better understanding of the function of genes associated with drug absorption and disposition. They claim that genome-scale CRISPR–Cas9 knockout screen study also facilitates the identification of novel genes in which loss alters drug permeability across biological membrane and thus modulates the efficacy and safety of drugs. Compared with conventional heterogeneous expression models or other genome editing technologies, CRISPR-Cas9 gene manipulation techniques possess significant advantages, including ease of design, cost-effectiveness, greater on-target DNA cleavage activity and multiplexing capabilities, which makes it possible to study the interactions between membrane proteins and drugs more accurately and efficiently.
Authors also provide an opinion that many mechanistic questions and challenges regarding CRISPR-Cas9 gene editing are yet to be addressed, ranging from off-target effects to large-scale genetic alterations.
Finally, authors conclude that the exploitation of CRISPR-Cas9 techniques has generated novel insights into the genes/proteins associated with biological barriers to drug delivery and may lead to the development of promising strategies to overcome therapeutic failures and minimize drug toxicity.
===========
Other comments:
1) Please check for typos throughout the manuscript.
2) Lines 492-504. Authors are kindly encouraged to cite the following article that describes therapeutic editing of p53 oncosuppressor gene using CRISPR/Cas9 system. DOI: 10.3390/genes11060704
Overall, the manuscript is highly valuable for the scientific community and should be accepted for publication.
Author Response
Ji He and colleagues present a quality and well-written review manuscript describing application of CRISPR-Cas9 system to study biological barriers to drug delivery.
Authors provide a thorough overview of the mechanisms of the widely used CRISPR-Cas systems in mammalian genome editing, as well as the challenges that restrict editing efficiency and the application of CRISPR-Cas9 in studying barriers to drug delivery.
Authors suggest that CRISPR-Cas9 has benefited the proof-of-principle studies for a better understanding of the function of genes associated with drug absorption and disposition. They claim that genome-scale CRISPR–Cas9 knockout screen study also facilitates the identification of novel genes in which loss alters drug permeability across biological membrane and thus modulates the efficacy and safety of drugs. Compared with conventional heterogeneous expression models or other genome editing technologies, CRISPR-Cas9 gene manipulation techniques possess significant advantages, including ease of design, cost-effectiveness, greater on-target DNA cleavage activity and multiplexing capabilities, which makes it possible to study the interactions between membrane proteins and drugs more accurately and efficiently.
Authors also provide an opinion that many mechanistic questions and challenges regarding CRISPR-Cas9 gene editing are yet to be addressed, ranging from off-target effects to large-scale genetic alterations.
Finally, authors conclude that the exploitation of CRISPR-Cas9 techniques has generated novel insights into the genes/proteins associated with biological barriers to drug delivery and may lead to the development of promising strategies to overcome therapeutic failures and minimize drug toxicity.
===========
Other comments:
- Please check for typos throughout the manuscript.
Ans: This gas been done accordingly.
2) Lines 492-504. Authors are kindly encouraged to cite the following article that describes therapeutic editing of p53 oncosuppressor gene using CRISPR/Cas9 system. DOI: 10.3390/genes11060704
Ans: This article has been cited in line 147.
Overall, the manuscript is highly valuable for the scientific community and should be accepted for publication.
Ans: Thank you for your kind comments.
Reviewer 3 Report
He et al. summarized the CRISPR-Cas9 system and its applications in drug delivery. Basic information about the CRISPR-Cas9 system, including protein structures, classification, PAM features, and factors that can affect editing efficiency, is well explained. Furthermore, the system’s applications in drug delivery are outlined. Although this review is very interesting, I have several concerns.
Concerns
- The title of this review is “Application of CRISPR-Cas9 system to study biological barriers to drug delivery.” However, the explanation regarding the system’s applications in drug delivery is extremely lacking compared to the explanation of basic information. The basic information, such as the class and subtypes, Cas9 orthologs, and PAM, could be more briefly summarized. The importance of the basic science is well understood, and indeed, the contents are very interesting. However, please consider the balance in terms of information presented inconsideration of the title.
- The authors mentioned CRISPR screenings in the applications in drug delivery; however, the basic concept of libraries is not well described. That would be helpful for readers who are not familiar with CRISPR.
- ABC transporters are well known as key regulator of drug resistance. Additionally, other regulatory transporters for drug delivery can be identified with CRISPR screenings (PMID: 33879555, PMID: 31523835). Therefore, information about recent findings using CRISPR techniques are also expected.
Author Response
He et al. summarized the CRISPR-Cas9 system and its applications in drug delivery. Basic information about the CRISPR-Cas9 system, including protein structures, classification, PAM features, and factors that can affect editing efficiency, is well explained. Furthermore, the system’s applications in drug delivery are outlined. Although this review is very interesting, I have several concerns.
Concerns
- The title of this review is “Application of CRISPR-Cas9 system to study biological barriers to drug delivery.” However, the explanation regarding the system’s applications in drug delivery is extremely lacking compared to the explanation of basic information. The basic information, such as the class and subtypes, Cas9 orthologs, and PAM, could be more briefly summarized. The importance of the basic science is well understood, and indeed, the contents are very interesting. However, please consider the balance in terms of information presented inconsideration of the title.
Ans: Many thanks for the constructive comments. The review manuscript has been restructured to balance the basic science and its application.
- The authors mentioned CRISPR screenings in the applications in drug delivery; however, the basic concept of libraries is not well described. That would be helpful for readers who are not familiar with CRISPR.
Ans: Thank you for the suggestion. The basic concept of CRISPR libraries have been added.
- ABC transporters are well known as key regulator of drug resistance. Additionally, other regulatory transporters for drug delivery can be identified with CRISPR screenings (PMID: 33879555, PMID: 31523835). Therefore, information about recent findings using CRISPR techniques are also expected.
Ans: Thank you for the constructive suggestion. CRISPR-screen identified contribution of SLC transporters for drug delivery together with recent findings have been added in the new section 3.2.2. and 3.2.3.
Round 2
Reviewer 1 Report
The authors have responded to the previous reviews and have removed several sections of text and reorganized the review.
Reviewer 3 Report
All concerns are well addressed.
Minor points;
Line 178,196, 204, 257, etc., the description of “P-gp (MDR1/ABCB1) ” should be unified through the manuscript.